# Factors Influencing Accounting Outsourcing Using the Transaction Cost Economics Model

Ivana Tomašević [1], Sandra Đurović [1], Nikola Abramović [1], Lidija Weis [2] and Viktor Koval [3,*]

1   Faculty of Business Economics and Law, "Adriatic" University, 16 Rista Lekića, 85000 Bar, Montenegro
2   Ljubljana School of Business, Tržaška cesta 42, SI-1000 Ljubljana, Slovenia
3   Department of Business and Tourism Management, Izmail State University of Humanities,
     68601 Izmail, Ukraine
*   Correspondence: victor-koval@ukr.net

**Abstract:** This paper presents the results of research conducted to identify the factors that influence the decisions of company management to outsource accounting services. A transaction cost economics (TCE) model was used to analyse factors that influence high levels of outsourcing of accounting tasks in the case of Montenegro, where, based on our sample, 75.4% of companies enter outsourcing arrangements with bookkeeping agencies or with external accountants. With an adaptation of International Accounting Standards (IAS) and legal requirements for submission of standardised year-end reports, there is evident growth of bookkeeping and financial service providers on the market and an evident trend of companies entering accounting service outsourcing with those agencies. A survey was developed to investigate 12 variables that, according to the TCE model, influence outsourcing decisions. The selection of variables was based on previous research in the field using the TCE or Resource-Based View model (the most common models used for this analysis). In contrast, new variables were introduced that measure the effects of the introduction of IAS through legal reporting obligations in Montenegro. By developing the model this way, it became possible to predict 47% of the variance of the dependent variable and to identify the main factors (other than price) that influence the decision of managers to outsource accounting services.

**Keywords:** accounting; financial reporting; outsourcing; transaction cost economics

## 1. Introduction

According to the theory of resource optimization in order to survive on the competitive market, companies need to reduce costs and optimize the use of their own resources (Plenert 1993; Wernerfelt 1984). Research shows that the main causes of enterprise failure are lack of planning, lack of knowledge and skills, lack of skills in the field of management and lack of key competencies (Dyer and Ross 2008). The following question arises: how do companies overcome the above shortcomings? One approach is reliance on outsourcing, i.e., engaging external service provider. In this way, companies can acquire knowledge and skills that they are currently lacking. This is particularly important for micro, small, and medium enterprises because they get the opportunity to improve their capacities, which allows them to become competitive and survive in the modern market (Anderson and McKenzie 2022).

The process of outsourcing starts with companies making the decision to enter the outsourcing process because they want to acquire lacking knowledge, skills, and competences in the market at a lower cost than what it would cost them to develop those services themselves (Espino-Rodríguez and Padrón-Robaina 2006). The theory of transaction cost economics (TCE) is based on this principle.

Entering the outsourcing process would reduce various costs for the company, such as salary, employment, and training of a new person who would need to learn to perform new tasks. Based on the TCE method, the company evaluates whether a particular job is

more favourable to outsource or to organize internally by choosing an alternative with lower costs.

The aim of this research is to understand factors that influence high levels of outsourcing of accounting in Montenegrin companies. A very high percentage of small and medium companies are outsourcing accounting; in particular, they are outsourcing the service of preparing year-end reports. Montenegro is a small economy with 99% of businesses being in the category of small and medium enterprise; therefore, in using TCE methodology we are bringing to light the main factors that impact accounting service outsourcing, which is very high (based on research 75.4%) and not elastic in terms of price changes. The same practice of outsourcing bookkeeping and accounting services exists in all Western Balkan countries; therefore, this research contributes to further research in the Balkan region.

### 1.1. Theoretical Framework

Empirical research related to outsourcing accounting is scarce. Several studies have been conducted to investigate the impact of the outsourcing of accounting advisory services on the business results of small and medium-sized enterprises (Kamyabi and Devi 2011; Bennett and Robson 1999). There are also a few studies on the outsourcing of accounting jobs, primarily in developed countries: Everaert et al. (2007, 2010); Doran (2006); Carey et al. (2006); and later in developing countries: Mashayekhi and Mashayekh (2008); and Jayabalan et al. (2009). However, there is no research on accounting services outsourcing in developing countries such as Montenegro, where, in the process of entering the European Union, there are new dimensions and factors that are influencing the outsourcing decision and that need to be taken into account. These factors are coming from the accession process and from the adaptation of the accounting system to European Union requirements with the introduction of International Accounting Standards and with under-developed capacities of SMEs in Montenegro to adapt to International Standards for Financial Reporting (ISFR).

For companies that 'do not have experience in accounting, it is very difficult to get familiar with and adapt to the generally accepted accounting principles, especially considering that the adaptation process requires much tacit knowledge. With accounting service outsourcing, a company can gain value because it gets quality service from an experienced and knowledgeable external accountant (Kamyabi and Devi 2011). By engaging an external agency or an individual with accounting knowledge, experience, and competence, the company can also reduce costs by reducing errors, improving performance monitoring, and improving productivity (better quality decision-making based on better information), which ultimately leads to better financial results (Gilley et al. 2004). These are the reasons why many companies are choosing to focus on their core business activities and outsource supporting accounting services (Weimer and Seuring 2009).

Empirical research on the outsourcing of accounting functions has given various insights that have helped improve our understanding in this area. In a survey of SMEs in Belgium, Everaert et al. (2007) concluded that companies prefer to combine outsourcing with internal accounting; they also identified asset specificity and frequency of accounting tasks as the most important factors that the company takes into account when deciding on the outsourcing. This has significantly contributed to understanding the need to outsource accounting, as well as to identify the main aspects of accounting outsourcing.

Furthermore, the studies carried out by Strouhal et al. (2010) pointed to the need for users of accounting reports to implement and improve accounting regulations for small and medium-sized enterprises.

The response to these problems was the creation of International Accounting Standards for Small and Medium Enterprises (IAS for SMEs), which reduced the disclosure requirements for SMEs as well as the financial and organizational burdens caused by reporting needs (Seifert and Lindberg 2010). Another important breakthrough in this area was the development of professional accounting services.

Accounting service providers have developed numerous skills and competencies, not just in administering accounting jobs (Nandan and Ciccotosto 2014), but also in management of knowledge and human resources (Parwita et al. 2021).

Research on accounting outsourcing also considers issues of additional knowledge and innovation developed through knowledge sharing in the outsourcing process. The main characteristic of the outsourcing process is that it involves subcontracting, transferring responsibility, and trust (Galloway et al. 2005; Espino-Rodríguez and Padrón-Robaina 2004; Espino-Rodríguez and Padrón-Robaina 2004). There are various reasons for which companies decide to enter the outsourcing process. In the United States, companies are entering this process in order to enhance quality of service that they themselves provide to a client. In Great Britain, on the other hand, the main reason for the outsourcing of accounting is to achieve the economy of scale. While in Asia, companies are looking to achieve higher specialization, therefore, in some cases, they outsource even core activities (Kakabadse and Kakabadse 2002; Zarrella and Huckhai 2004).

*1.2. Definition of the Problem*

The problem we researched was that of identifying the factors that most influence the company's decision on a complete or partial outsourcing of accounting services. The research exposes problems encountered by companies that neither have internal accounting nor have the capacity to develop an internal accounting department on their own. An additional area we explored is whether adapting to international professional accounting regulations has an impact on outsourcing, i.e., if adopting International Accounting Standards with simplified reporting procedures in the form of standardized end-of-year reports and reduced reporting obligations, that depend on the size and the income of companies, is influencing a growth or decrease in outsourcing in accounting.

Depending on the capacity that a company has, the management of the company can decide to outsource accounting tasks, either partially or completely, to an external accountant or to a company that is specialized in providing accounting services (external accounting agencies).

On the other hand, a company can have an internal accountant or accounting department and conduct all accounting tasks internally (Chorna et al. 2019). The topic of the research is to understand the factors that influence the decision of management to use external accounting agencies for providing basic accounting services.

For research purposes, identification of basic accounting tasks that companies perform during the year was conducted and the tasks were classified in groups (Everaert et al. 2010):

1.    Recording and tracking invoices and accounting transactions.
2.    Preparing monthly or quarterly accounting reports.
3.    Conducting specific calculations such as amortization, taxes, etc.
4.    Preparing end-of-year reports: based on International Financial Reporting Standards (IFRS), these reports are standardized and contain the Balance Sheet, Income Statement, and Statistical Annex in standardized forms.

All the above-stated tasks are considered basic tasks, while the advanced financial tasks such as auditing, ratio analysis, or similar ones were not part of our research as they were influenced by additional factor.

## 2. Methodology

The methodology that was used to research factors that influence accounting service outsourcing was transaction cost economics (TCE). Transaction cost economics suggests that expenses and difficulties associated with market transactions may, in some cases, favour internal organization of service (in house services or production) and, in other cases, favour outsourcing (acquiring required services or product externally on market and under market conditions) (Gilley et al. 2004).

The decision of whether certain tasks will be conducted internally or will be outsourced depends on the transaction expenses of both alternatives. Company management will select the alternative with lower transaction expenses (Espino-Rodríguez et al. 2008).

The transaction cost of accounting outsourcing (other than the cost of engaging an external agency or external accountant) must include the costs of researching and the contracting process with the accounting agency as well as the costs of monitoring and feedback (Williamson 1985). Based on the TCE, outsourcing is preferred in situations where the market is open and when there are many service providers that are offering accounting services. The competition that exists in this case between accounting agencies and external accountants as providers of accounting services will reduce the need for monitoring their behaviour. On the other hand, in market conditions where there are a limited number of providers of accounting services and where market mechanisms such as competitive pressures do not function, it becomes very "expensive" to engage an external accountant, as their behaviour must be monitored, the contractual relationship must be strengthened, and constant control has to be carried out. In this case, based on TCE, it is more convenient to replace an external accountant with a company's employees (Hafeez and Andersen 2014; Edeh et al. 2023).

The TCE method (Coase 1991) has become a standard framework for assessing why an enterprise performs some activities internally and outsources others. When talking about accounting outsourcing (Everaert et al. 2010), four important factors that affect accounting are identified:

1. Specificity of assets.
2. Uncertainty of the environment.
3. Uncertainty surrounding the behaviour of accounting agencies.
4. Frequency of accounting tasks.

This research is based on the testing of hypotheses related to those four factors and includes two new factors: trust in external service providers by Montenegrin companies and the influence of International Accounting Standards on the market.

The research used was as follows: dialectical method, content analysis, statistical method, comparative method, and induction method.

*2.1. Data Collection*

The survey was conducted based on data from the questionnaires which were filled in by: the managers of the company or by the person in charge of accounting, (i.e., the financial director or accountant). After the questionnaire, interviews were also conducted with owners and directors of accounting agencies, independent accountants, auditors, and other providers of accounting services.

A random sample (without repetition) of 200 companies was prepared for the needs of the research (based on data from the Central Register of the Commercial Court). Enterprises were distributed according to the sectors in which they operate in order to adequately reflect the enterprises and to perform cross-sectoral research. In the selection of the sample, the accounting agencies were excluded due to the nature of the work, which is specific and could affect the results. An accounting agency's outsourcing refers to the transfer of a part of the bookkeeping job to third parties (an accountant or another agency).

In this case, three entities are involved in the relationship: a company, an accounting agency, and an agency to which a portion of the tasks are outsourced. This outsourcing is specific and complex, since, apart from outsourcing between the company and the accounting agency, outsourcing between the accounting agency and a third entity must also be observed. Because of the complexity of these relationships, accounting agencies are excluded from the analysis. The sample covers sectors where companies in Montenegro are most often represented: trade, services, tourism and hotel industry, pharmacy, manufacturing, construction, and information technology. SMEs comprise 99.5%, or 25,991, of companies in Montenegro, which is also reflected in the sample.

The questionnaire was answered by 126 companies, which represented 63% of the selected sample; therefore, the survey was successfully implemented because the percentage of response was very high. The size of the sample is comparable with studies in which SMEs have been surveyed with a sample of 100 to 180 companies.

*2.2. Description of Independent Variables and Hypotheses Used in the Research*

Independent variables are factors that influence the company's outsourcing accounting. There are 5 factors that influence company's decision about outsourcing accounting and those are:

1. Specificity of assets.
2. Uncertainty in the environment.
3. Uncertainty of behaviour.
4. Frequency of performing accounting tasks.
5. Trust.

2.2.1. Specificity of Assets Invested in the Company's Resources

In accordance with transaction cost economics, one of the main factors affecting the demand for the outsourcing of accounting services, and the intensity of outsourcing, is the specificity of assets (McIvor 2009; Kamyabi and Devi 2011). Some goods or services can be produced more efficiently if one or both actors invest in "transaction-specific" assets, which are not easy to put to another use if the relationship between the companies involved is interrupted. Because of this, the specificity of the assets is significant from the standpoint of the economy of transaction costs. The specificity of assets is the degree to which assets can be used in several situations and goals. An asset with a high degree of specificity is used only in certain situations or for specific purposes. An asset with medium or low specificity has multiple uses and purposes. Depending on the type of asset, the specificity of the funds can be related to, for example, a workforce trained to perform only one task, and it has limited use due to some inherent limitations for other possible uses. There are five basic forms in which the specificity of funds can arise, which are:

1. Specificity of equipment or tools, e.g., equipment and machines that produce inputs for specific clients or are specialized to use the input of a particular supplier.
2. Location specificity refers to the fact that the funds invested in the site are not mobile and cannot be reoccurred in any way.
3. Specificity of investment in human capital refers to the accumulation of certain knowledge and expertise that is particularly important for the enterprise.
4. Assets for specific needs involve investment in assets for the specific needs of a certain client, and if this client decides not to order from the company anymore, there will be excess capacity in the company because equipment and tools cannot be easily transferred for other use.
5. Specificity of brand or reputation occurs in the cooperation between companies when the common brand reputation must be maintained. It is particularly characteristic for the franchise relationship—the reputation of the entire franchise will depend on the behaviour of each franchisor.

From the above classification, it is possible to conclude that, according to the specifics, funds can be classified as tangible and inviolable assets. From the point of view of accounting, touchable assets are, for example, accounting software, while untouchable assets are related to human capital, such as information, knowledge, and competencies of accountants. Knowledge, i.e., human capital, is specific when professional accountants include specific skills to solve a business-related issue and thus provide a special (specialized) accounting service (Everaert et al. 2010). In accordance with the economics of transaction costs, where transactions are routine and the specificity of funds is low, such transactions are more often outsourced (Chang et al. 2009; Jiang et al. 2007; Watjatrakul 2005). In accordance with the TCE, when it comes to high specificity of assets requiring specialized accounting knowledge,

a search for an adequate accounting agency that can provide specialized services is longer, and the procedure for contracting with the agency is more complicated, and therefore such activities will be less outsourced (Nicholson et al. 2006; Espino-Rodríguez et al. 2008).

In accordance with the above, if accounting activities are significantly adapted to the company's needs and the specificity of the funds is growing, the transfer of accounting functions to a professional agency will be more problematic and costly. Everaert et al. (2010) and Kamyabi and Devi (2011), in research conducted in Belgium, showed that there is a great correlation between the specificity of the asset and the outsourcing of accounting, and, therefore, this factor is further analysed in the literature. Regardless of methodology used (the two most common methods used to explain outsourcing of accounting services are TCE or a resource-based model), the specificity of the assets is taken as a factor that influences the outsourcing, which was performed in this analysis by presenting the following hypothesis:

**H1.** *There is a negative and significant relationship between specificity of resources and the outsourcing of the accounting tasks.*

### 2.2.2. Uncertainty in the Environment

Uncertainty in the environment when considering accounting and auditing services refers to expected variations in demand for accounting services that are based on stability and predictability in relation to the amount of accounting work (Widener and Selto 1999) which is again a consequence of the uncertainty of business activities. For example, if business activities are uncertain and changeable, resulting in an unstable number of invoices, orders, seasonal contexts, changes in trends, etc., then the amount of activity to be performed by the accounting agencies is uncertain. This is also typical for cases of mergers and acquisitions, the closing of businesses, fast-growing companies, etc. Under these conditions, the costs of accounting services are higher because they cannot be accurately estimated and foreseen, and there is a possibility that the contract with an external accountant must be audited, changed, that additional services will have to be paid, etc.

On the other hand, when an enterprise can predict the scope of the work, accounting tasks will be easily predicted, and the company will be able to outsource accounting with low transaction costs. A high level of uncertainty also increases the transaction cost of preparing and strengthening contracts with external accountants. In a contract, as many conditions as possible need to be defined and assessed. Under conditions of uncertainty, an internal accountant will be able to respond to fluctuations more quickly, and hence, in such situations, the internal organization of the accounting function is preferred (Hennart 1994). This leads to the following hypothesis:

**H2.** *There is a negative and significant relationship between uncertainty in the environment and the outsourcing of the accounting task.*

### 2.2.3. Uncertainty in Behaviour

The term opportunism or uncertainty in the behaviour of accounting agencies affects business results in relationships between a company and an accounting agency. According to research the opportunism of the agency is reflected in "failures to distribute, distort, disclose, or otherwise act un-conceitedly and fraudulently for the purpose of the agency's own profit" (Chen et al. 2002; Wang et al. 2023). Opportunism can occur before the transaction occurs (ex-ante) or while the transaction lasts (ex-post). Ex post uncertainty is harder to spot and control before it happens (Lai et al. 2012). Although opportunism is widespread in various relationships and has been researched in the literature on various transaction costs, only a few studies have dealt with ex-post opportunism in accounting transactions and how it affects the relationship between a company and an accounting agency (Jap and Anderson 2003). TCE theory shows a positive correlation between the level of dependence on the partner in the transaction and the tendency for the partner to act in opportunistic and uncertain ways (Hawkins et al. 2009). Opportunist behaviour reduces trust, commitment, cooperation, satisfaction, and the long-term relationship between

businesses and providers of accounting services (Lee et al. 1997). According to Everaert et al. (2010), if external accountants develop a relationship based on trust, transparency of information, and predictability of behaviour, this will minimize opportunism and, therefore, according to TCE theory, companies will be willing to enter an outsourcing agreement with an accounting agency:

**H3.** *There is a negative and significant relationship between uncertainty in the behaviour of professional agencies (accountants, accounting agencies) and outsourcing of the accounting tasks.*

2.2.4. Frequency of Accounting Tasks

The frequency of conducting accounting tasks is closely linked with the asset specificity factor, and it is one of the main factors influencing an outsourcing decision. Frequency refers to the amount of similar accounting transactions to be performed within a company. A high frequency of transactions that are similar in nature can lead to the economy of scale advantages that will be enough to compensate the expenses of setting up accounting services within a company (Everaert et al. 2010; Yankovyi et al. 2021). According to TCE theory, if a company needs to perform certain accounting tasks frequently, or performs repetitive accounting tasks similar in nature frequently, then it is more likely that the company will be willing to organize internal accounting and less likely to enter the outsourcing process with an external service provider.

**H4.** *There is a negative and significant relationship between frequency of accounting tasks and the outsourcing of accounting tasks.*

2.2.5. Trust in External Service Providers

Trust in external service providers is important in deciding to outsource accounting services. Although this variable was not used in previous research on this topic, there is a sound theoretical basis for its inclusion as an independent variable. From the TCE model, it is implied that if there is greater trust among the actors in the process of outsourcing accounting, there is less need to establish a mechanism for controlling the accountant's behaviour, and therefore the costs of outsourcing are lower. Trust is therefore closely related to behavioural uncertainty (opportunistic behaviour of external service providers). Accounting is a highly regulated field, with laws on accounting and auditing, International Accounting Standards, and high fees for not complying with the law. This regulation reduces the opportunistic behaviour of actors in the market and increases trust in sharing knowledge and information through the outsourcing process. Therefore, we can conclude that:

**H5.** *There is a positive and significant relationship between trust in the professional work of an accountant (accounting agencies) and the outsourcing of accounting tasks.*

*2.3. Description of Control Variables Used in Research*

Control variables are those that are kept constant in the experiment to be able to observe the relationship between the other variables. Control variables are not of primary importance for testing and do not affect the result, but any change in their values can lead to a correlation between the dependent variable and the independent variables.

Research shows that the use of accounting outsourcing has a positive correlation with the age and size of the company (Bennett and Robson 1999; Dyer and Ross 2008). Because of this, these two parameters were taken as control variables. This was similarly performed in the Gooderham et al. (2004) and Everaert et al. (2010) studies.

Previous studies have also shown that management education influences the decision on outsourcing, i.e., there is a positive correlation (Park and Krishnan 2001; Everaert et al. 2010). Because of the above, in order to control the impact of education, such as in the studies carried out by (Everaert et al. 2010) and (Kamyabi and Devi 2011), the respondents were asked to provide a qualification level as well as to identify whether they a University degree in the field of economics (because it is closely related to accounting).

Also, due to the specificity of the accounting work, as a control variable, it was necessary to include the differentiation between service and production companies. Service companies have less asset specificity than, for example, construction companies; additionally, the types of accounting jobs that are outsourced by service companies are considerably simpler, and therefore service companies have a positive correlation with accounting outsourcing (Everaert et al. 2010).

The last factor involved as a control variable was the gender of the respondents. Research shows that gender can influence the decision to outsource services (Howcroft and Richardson 2008; Cohen 2005). Since the 1970s, there has been a tendency to employ more women in bookkeeping (Cooper and Taylor 2000).

The research covers the outsourcing of a complete accounting service rather than specific accounting issues (i.e., accounting areas requiring specific knowledge such as mergers, acquisitions, intellectual property rights, etc.), and so the impact of gender on the respondents must be observed. The assumption is that women would rather perform accounting tasks than outsource them.

The last control variable that was introduced in the analysis was the degree of firm accounting reporting obligations. According to the Accounting Law, most small and medium enterprises in Montenegro are obligated to prepare end-of-year reports at the end of each business year. Those reports need to be prepared in accordance with IAS (International Accounting Standards) and ISFR (International Standards for Financial Reporting).

However, due to a company's size and income considerations, some companies have additional reporting obligations. Those additional reports are the management report, the audit report, and the report about payments to the state or local authorities. The assumption is that the more reporting the company needs to provide, the more specialized services and accounting knowledge they will need; therefore, they will need to subcontract (outsource) an external accountant.

### 2.4. Linear Regression Model Used in the Analysis

The interdependence of the dependent variable (outsourcing of accounting) and independent variables (specificity of assets, uncertainty in the environment, uncertainty in the behaviour of the accounting service provider, frequency of accounting tasks, and trust) were examined using the following linear regression model:

$$\text{OA} = a_1 + a_2 + a_3 + a_4 + a_5 + a_6 + a_7 + a_8 + a_9 + a_{10} + a_{11} + a_{12} + a_{13} \tag{1}$$

In this linear regression model variables have following meaning: OA is outsourcing accounting; $a_1$ is assets; $a_2$ is accounting reporting obligations; $a_3$ is university degree; $a_4$ is degree in the field of economics; $a_5$ is age of the firm; $a_6$ is size of the firm; $a_7$ is firm field of work (service firm); $a_8$ is trust; $a_9$ is asset specificity; $a_{10}$ is uncertainty in the environment; $a_{11}$ is uncertainty of behaviour; $a_{12}$ is frequency of accounting task; $a_{13}$ is gender of respondents.

Data processing and analysis was performed using the Statistical Package for Social Sciences (SPSS). SPSS Statistics is a statistical software suite developed by IBM, United States of America, for data management, advanced analytics and multivariate analysis.

## 3. Results

### 3.1. Descriptive Statistics

The survey, which was conducted inMontenegro included 126 companies registered in the Central Register of the Commercial Court. The sample was representative with a confidence interval of 7.54. A confidence interval of 7.54 percent implies 95% confidence rating (95–7.5) and % (75 + 7.5). The number of companies in Montenegro according to national statistics presented in MONSTAT amounted to 30,286 in 2017. Compared to 2016, the number of active companies increased by 7.8%. Our research used the following methods: dialectical method, content analysis, statistical method, comparative method, and induction method. During the data analysis, the statistical software SPSS was used.

The average age of the company is between 5 and 10 years, which is recorded as category 3 in the SPSS program and detailed in Table 1.

**Table 1.** Enterprises in relation to the number of years in operation, (n 126).

| Number of Years in Operation | | Frequency | Percentage | Valid Percentage | Cumulative Percentage |
|---|---|---|---|---|---|
| **Valid** | Less than 2 years | 11 | 8.7 | 8.7 | 8.7 |
| | 2–5 years | 24 | 19.0 | 19.0 | 27.8 |
| | 5–10 years | 32 | 25.4 | 25.4 | 53.2 |
| | 10–20 years | 32 | 25.4 | 25.4 | 78.6 |
| | more than 20 years | 27 | 21.4 | 21.4 | 100.0 |
| | **Total** | **126** | **100.0** | **100.0** | |

Average number of employees was from 0 to 49 employees as presented in Table 2.

**Table 2.** Enterprises in relation to number of employees, (n 126).

| | | Frequency | Percentage | Valid Percentage | Cumulative Percentage |
|---|---|---|---|---|---|
| **Valid** | From 0 to 49 | 101 | 80.2 | 80.2 | 80.2 |
| | From 50 to 250 | 15 | 11.9 | 11.9 | 92.1 |
| | Over 250 | 10 | 7.9 | 7.9 | 100.0 |
| | Total | 126 | 100.0 | 100.0 | |

In Montenegro, 99% of the companies are small and medium enterprises, (SMEs) and in the sample used in the research 92% of the companies are SMEs (Table 3).

**Table 3.** Statistical review of the sample in relation to the control variables, company year and number of employees.

| | | Company Age | Number of Employees |
|---|---|---|---|
| | Valid | 126 | 126 |
| | Missing | 0 | 0 |
| Arithmetical mean | | 3.32 | 1.28 |
| Standard error | | 0.111 | 0.054 |
| Median | | 3.00 | 1.00 |
| Mode | | 3a | 1 |
| Standard deviation | | 1.250 | 0.602 |
| Variance | | 1.562 | 0.362 |
| Range | | 4 | 2 |
| Minimum | | 1 | 1 |
| Maximum | | 5 | 3 |
| Total | | 418 | 161 |
| Percentage | 25 | 2.00 | 1.00 |
| | 50 | 3.00 | 1.00 |
| | 75 | 4.00 | 1.00 |

### 3.2. Regression Analysis

The basics of multiple linear regression are to evaluate whether the dependent variable can be predicted from a set of independent (or predictive) variables. Or, in other words, how many variations in dependent variables are coming from the impact of independent variables on control variables? A specific approach to regression selection is helpful in testing the predictor, which, in the end, increases the efficiency analysis (Table 4).

For research purposes, our accounting outsourcing variable is defined as a binary variable where all companies (business entities) that have identified partial outsourcing were treated as companies that have entered the outsourcing process (Table 5).

**Table 4.** Items used for measuring Accounting Reporting Obligations.

| | **For measuring Accounting Reporting Obligations According to Law on Accounting** | |
|---|---|---|
| RN 1 | According to Law on Accounting our company is obligated to prepare year-end reports on December 31, at the end of the business year. These reports need to be prepared in line with IAS and ISFR | Questions are adapted from Law on Accounting and introduced as new variable to measure |
| RN 2 | According to Law on Accounting our company needs to submit year-end reports to the Tax authority in Montenegro that contains: Balance Sheet, Income Statement and Statistical Annex | |
| RN 3 | According to Law on Accounting our company needs to submit management reports together with year-end reports | |
| RN 4 | According to Law on Accounting our company needs to submit management reports and also a report about payments to the state or local authorities together with year-end reports | |
| RN5 | Our company is in obligation to submit an audit report together with all above stated reports | |

**Table 5.** Items used for measuring outsourcing of accounting services.

| **SPSS Code** | **Items** | **Source** |
|---|---|---|
| OAS 1 | Internal Accounting | Question is adopted according to research goal |
| OAS 2 | Accounting Outsourcing | |

Based on the hypotheses and goals of the research final model was presented in Figure 1.

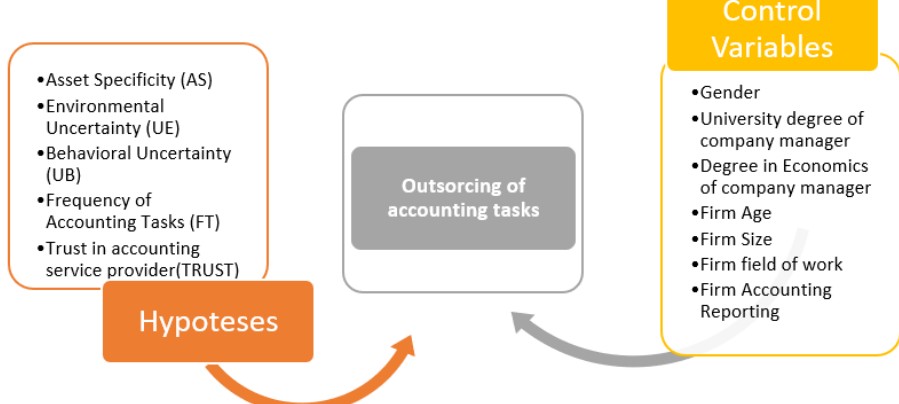

**Figure 1.** Visual presentation of research hypotheses.

The numerical data about basic characteristic of sample are presented in Table 6.

In Table 7, the variables that were researched are: TRUST—trust in the accounting service provider; AS—asset specificity; UE—environmental uncertainty; UB—behavioural uncertainty; FAT—frequency of accounting tasks; ARO—firm accounting reporting obligations.

Table 8 shows the correlation analysis of the variables used in the model: regression coefficient (B), Wald statistics (for testing statistical significance) and the relationship Ratio CI for (EXP (B)) for each category of variables.

By observing the results, we can first notice the significance of gender for the outsourcing of accounting services, and there is a significant overall effect (Wald = 4.781, df = 1, Sig.

$p$ =< 0.05). Coefficients (B) for gender are significant and negative (the reference is a female gender), which suggests that men are expected to have a greater chance of engaging an external service provider for providing accounting service. More precisely, in accordance with Exp (B) = 1 − 0.038 = 0.962 or 96.2% in relation to females.

**Table 6.** Basic characteristic of sample.

| N = 126 | Sample |
|---|---|
| Gander of company manager | |
| Female | 44.4% |
| Male | 55.6% |
| University degree of manager | |
| Yes | 46.8% |
| No | 53.2% |
| University degree in Economics of manager | |
| N = 59 | |
| Yes | 61.1% |
| No | 38.9% |
| Firm's age | |
| Less than 2 years | 8.7% |
| From 2 to 5 years | 19.0% |
| From 5 to 10 years | 25.4% |
| From 10 to 20 years | 25.4% |
| Above 20 years | 21.4% |
| Firm size | |
| 0–49 | 80.2% |
| 50–250 | 11.9% |
| More than 250 | 7.9% |
| Trust (scale from 1 to 5) | |
| N = 107 | |
| Medium Value | 4.78 |
| Standard deviation | 0.48 |
| Service company | |
| Yes | 92.9% |
| No | 7.1% |

**Table 7.** Medium value and standard error of all researched variables.

| Variable | Medium Value | Standard Error |
|---|---|---|
| TRUST | 4.780 | 0.480 |
| AS | 3.831 | 1.101 |
| UE | 2.341 | 1.332 |
| UB | 4.828 | 0.527 |
| FAT | 176.869 | 79.3 |
| ARO | 4.4352 | 0.804 |

**Table 8.** Correlation analysis of variables used in the model.

| | B | S.E. | Wald | df | Sig. | Exp(B) | 95% C.I. for EXP(B) | |
|---|---|---|---|---|---|---|---|---|
| | | | | | | | Lower | Upper |
| Gender | −3.276 | 1.484 | 4.871 | 1 | 0.027 | 0.038 | 0.002 | 0.693 |
| University degree | 0.116 | 0.939 | 0.015 | 1 | 0.901 | 1.123 | 0.178 | 7.070 |
| Degree in economics | 0.974 | 1.097 | 0.788 | 1 | 0.375 | 2.648 | 0.308 | 22.722 |
| Firm Age | −0.444 | 0.363 | 1.490 | 1 | 0.222 | 0.642 | 0.315 | 1.308 |
| Firm size (based on number of employees) | −1.060 | 0.677 | 2.453 | 1 | 0.117 | 0.346 | 0.092 | 1.305 |
| Firm's field of work (service or production) | 0.711 | 1.474 | 0.233 | 1 | 0.630 | 2.036 | 0.113 | 36.637 |
| TRUST | 1.893 | 0.867 | 4.772 | 1 | 0.029 | 6.639 | 1.215 | 36.286 |
| AS | −1.501 | 0.726 | 4.277 | 1 | 0.039 | 0.223 | 0.054 | 0.925 |
| UE | −0.458 | 0.317 | 2.084 | 1 | 0.149 | 0.633 | 0.340 | 1.178 |
| UB | 0.495 | 0.811 | 0.372 | 1 | 0.542 | 1.640 | 0.335 | 8.038 |
| FAT | 0.628 | 0.456 | 1.897 | 1 | 0.168 | 1.873 | 0.767 | 4.576 |
| ARO | 0.006 | 0.006 | 0.896 | 1 | 0.344 | 1.006 | 0.994 | 1.018 |
| Constant | −0.700 | 4.395 | 0.025 | 1 | 0.873 | 0.497 | | |

Furthermore, based on the results of the binary multiple linear regression, we have arrived at the conclusion that the regression coefficient (B) for the variable TRUST—confidence—is significant and positive. In other words, for each unit increase in the variability of the TRUST, the probability of outsourcing accounting services will increase. This prediction is statistically relevant at the level of 0.05. In other words, for each individual increase in the level of trust (confidence in the accounting service provider), a firm's decision to enter into accounting outsourcing is expected to increase by 1893, keeping all other independent variables constant. Additionally, the increase in the probability of outsourcing accounting activities is higher by a factor of 6.639 for each increase in the TRUST unit.

For each increase in one AS unit, the outsourcing of accounting services is reduced by 1501, and the probability is statistically significant at 0.05. For each 1 unit of increase in AS, the probabilities change by −0.777. The probability of outsourcing (engagement of an external accountant) is lowered by a factor of 0.777 for each unit of AS increase.

The model characteristics shown in Figures 1 and 2 show that the control variables of gender, trust, and the independent variable specificity of the assets of small and medium enterprises (AS of SMEs), have a statistically significant influence on the explanation of the variance of the independent variable, which is also shown in Table 7, where the result of the tested research hypotheses is summarized.

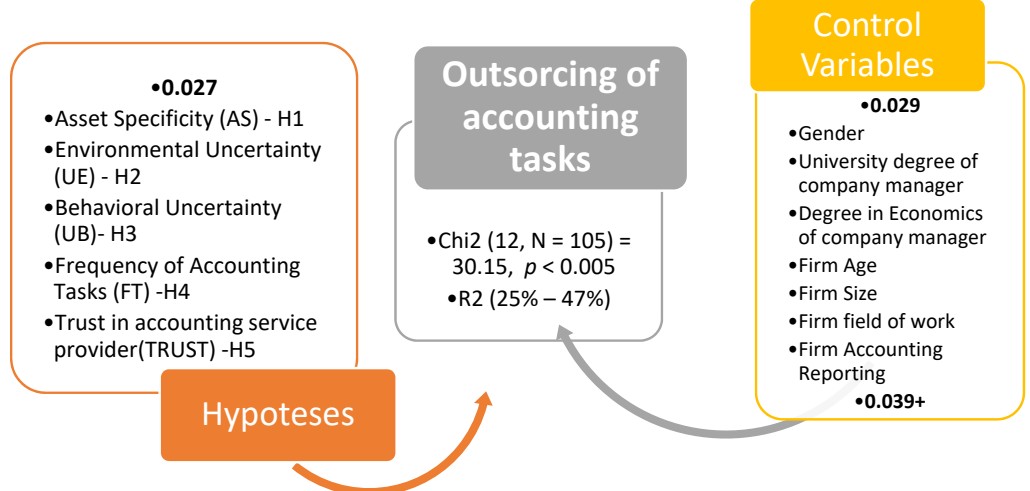

**Figure 2.** Presentation of the results of the research hypotheses.

## 4. Discussion

The results showed that 55.6% of companies use an external accounting service to outsource all accounting tasks, from the input of invoices to annual reporting. In other words, both routine and non-routine tasks are outsourced; 19.8% of companies are using partial outsourcing, where they only outsource portions of accounting tasks, mainly end-of-year reports and some specific calculations (amortization, closing the accounts, calculating revenue for tax purposes, or similar)—non-routine tasks are outsourced. Only 24.6% of the researched companies have internal accounting services and are not relying on accounting outsourcing in any part of their work. Our sample therefore of the researched companies in Montenegro showed very high reliance on accounting outsourcing from companies in Montenegro. Cumulatively, 75.4% of companies were entering some type of accounting outsourcing process and, compared to the results shown on samples made in different countries (EU countries and the United States of America), this percentage is very high. Compared to previous research where this percentage ranges from 14% to 30% (Everaert et al. 2010), we were able to highlight very high reliance of companies in Montenegro on Accounting Service providers.

The research has shown that outsourcing accounting supports the H1 and the H5 hypothesis of the TCE model. H1 is a hypothesis that tests the relationship between the specificity of the company's assets and the outsourcing of accounting. Regardless of whether or not companies are outsourcing this service, company representatives consider that an external accountant must be aware of the specificity of the company's assets in order to be able to engage in accounting work.

Companies with a high specificity of assets also must consider that if they decide to outsource accounting, it would be expensive to change the accountant. Because any new accountant would need to have same specific knowledge and be able to provide specialized accounting tasks. A moderate correlation is observed between the specificity of non-routine tasks and outsourcing accounting. In organizations where business knowledge of a particular context is needed for accurate bookkeeping (telecommunications, retail, ports, construction businesses, etc.), it becomes expensive for those skills to be outsourced; therefore, the company must develop internal accounting capabilities (units with specific skills).

This result is consistent with previous research of the topic conducted in Belgium (Everaert et al. 2010) where there are a smaller number of companies entering into outsourcing agreements. Research conducted in Pakistan also produced the same result, demonstrating that even on markets where there is high level of outsourcing accounting service, hypothesis of the TCE model H1 and H5 are supported (Hafeez and Andersen 2014). Regarding the specificity of the asset H1, it has also been proven that specific jobs and tasks continue to be performed internally, while standardized jobs such as the final reports are more often outsourced.

The existence of trust plays a vital role in testing the hypothesis and is presented specifically as H5. Our model confirmed a positive correlation between trust and the outsourcing of accounting. In the first phase of research using the Pearson coefficient of correlation, a very high linear correlation between outsourcing and trust was shown. Subsequently, in the regression analysis, H5 was proven. In previous research, trust was often used as a control variable because outsourcing relationships involve the transfer of knowledge and experience and, therefore, trust is involved in relationships (Gooderham et al. 2004).

Research shows that companies with an external accountant had a high degree of trust in external accountants and were therefore ready to engage in full or partial outsourcing (Glavee-Geo et al. 2022). A high correlation between uncertainty in behaviour and outsourcing was observed on the sample. Pearson's correlation coefficient showed a value of $p = 0.95$ with a probability ($\alpha = 0.05$) that there is sufficient evidence that there is a very good to excellent connection between the accounting outsourcing and the accuracy of the year-end report. Only one company in the complete sample stated that they do not trust the accuracy of the report. The entry of invoices, accounts, and financial transactions by

accounting agencies is perceived as accurate, which is a reflection of high trust in the work of accounting agencies and external accountants. Uncertainty in behaviour is a factor introduced in 2007 in the analysis by Vandaele et al. (2007) as a particularly significant factor in outsourcing of services; however, in research conducted on the topic of outsourcing accounting services, uncertainty in behaviour has not been identified as a significant factor through regression analysis. This is explained by the fact that, in the modern market with high level of digitalization, it is very easy to get information about the accuracy of the data that the accountant performs (Gulin et al. 2019). This information can be obtained primarily from the software used by the accountant, and then by the National Authority that collects the data (Everaert et al. 2010)—in Montenegro that is the National Tax Authority. Existence of this open and transparent control mechanism also affects market uncertainty.

The current economic climate in which companies operate in Montenegro is relatively stable, and companies do not perceive seasonal variation as significant. Under the existing conditions, the environment is not perceived as uncertain. On the other hand, lower uncertainty in the environment has influenced the greater outsourcing of accounting services, especially non-routine tasks. The survey conducted on the sample was not able to confirm H2 with this model. In conditions of low uncertainty in the environment and the introduction of standardization in accounting (standardized accounting end-of-year reports as well as adaptation of IAS), the accounting activities themselves, and especially non-routine tasks, reached a high level of outsourcing of as much as 75%. Therefore, it is easy to understand why this factor does not show a high correlation with outsourcing. Because Montenegro is in the process of gaining access to the EU market, businesses do not see uncertainty in the environment as a factor that explains or affects accounting authorisation. This is similar to research conducted in EU countries (Everaert et al. 2010) and different from research conducted in less stable countries (Hafeez and Andersen 2014). Another control variable that showed a high correlation with outsourcing is gender. Companies with female respondents (female managers and/or owners) are less likely to outsource accounting. Male managers were more likely to outsource accounting services. Again, we can relate this to the division of power and the fact that many earlier studies show that men are more likely to take risks and that women prefer control over risk. The risk exists in an outsourcing relationship as well as in other types of cooperation between businesses. When a company organizes an internal accounting service, there is more control over the work and therefore the risk is lower. Of course, this variable and the relationship between gender structure and outsourcing in general should be further explored. In practice, it has been observed that most accounting work is carried out by women, both internally in the company as well as within external accounting agencies. This phenomenon should be the subject of new research.

Authors should discuss the results and how they can be interpreted from the perspective of previous studies and the working hypotheses. The findings and their implications should be discussed in the broadest possible context. Future research directions may also be highlighted.

## 5. Conclusions

This research has undoubtedly shown that the outsourcing decision cannot be based solely on the characteristics of the transaction but also that there are certain other variables that influence the decision, such as trust, reporting obligations, gender structure, and specificity of the assets. The entire model used 12 variables to predict accounting outsourcing, and by testing the values of the pseudo-R squares "Cox & Snell R square" and "Nagelkerke R square" 25% and 47% of the variance of the dependent variable was explained and predicted from the independent variables used in the model. This percentage is very high compared to previous studies in which the value of the pseudo-R squared reached up to 30%. When it comes to accounting in Montenegro, often knowledge acquired through formal education is characterized as "theoretical", while accounting knowledge that is needed for running a business is considered to be "practical, tacit, and gained through

experience". This perception is very important for making an outsourcing decision. In an environment where there is such a high degree of outsourcing, a lack of knowledge about the specifics of accounting issues (cases, calculations, and reporting needs) could affect the decision on outsourcing. It is necessary to further investigate how much a lack of knowledge in the field of accounting reporting obligations affects outsourcing.

It is unambiguously shown in the sample that companies that have internal accounting better perceive their knowledge and abilities, as well as the accuracy with which they carry out accounting. They have clear knowledge about the obligations regarding the law in this area; they have completed the questionnaire that was used in the research more accurately. On the other hand, a large percentage of the representatives of the companies that outsource accounting do not have a clear picture of their accounting and tax obligations. It is possible that this affects their wrong perception of the difficulty of accounting tasks, lack of information, lack of understanding, and lack of interest. Another area of possible further research is how much a lack of information and wrong perceptions about the complexity of accounting tasks affect accounting outsourcing.

Another thing to consider is the low costs of services of accounting agencies. Recent trends show that change in price of up to 20% did not have any effect on companies to move from external service provider to internal accounting service; therefore, further research in the area of elasticity of accounting outsourcing to price changes is advised. This can be applied to the six Western Balkan countries, which have similar standardized accounting forms and have adopted International Accounting Standards (IAS) and are preparing reports according to International Financial Reporting Standards (IFRS).

Accounting agencies need to perform most of the tasks uniformly, and with economies of scale, they can perform those tasks at a lower cost. Additionally, accounting agencies build up tacit knowledge that is specific to professional service providers, which is very hard (and expensive) for companies outside of the accounting field to acquire. This market situation discourages companies' own engagement in acquiring accounting knowledge, and this area represents an important field for further research and analysis of such a high level of outsourcing. Regarding asset specificity, many respondents did not answer the question about accounting software, as most companies researched do not use specific accounting software. It would be necessary to explore whether there are adequate accounting software programs in Montenegro, how much the level of interdisciplinary knowledge (accounting, computer programs, foreign languages, etc.) is needed for the program to be used, and the extent to which there is the need for the adjustments of the reports that the system presents to standardized reports by IFRS and IAS.

When it comes to the frequency of accounting tasks, research in Montenegro showed that the frequency of accounting tasks has a weak effect on outsourcing decisions. In other research conducted in previous years, from 2006–2010, the situation was different, and if accounting tasks were more frequent, companies would not outsource them but keep internal accounting. However, with the development of standardized end-of-year reports, automatization and networking of transactions have become less important for accounting outsourcing. Cloud-based AIS provides easy access, scalability, integration, and, according to research, seems to encourage users of cloud-based AIS to outsource frequently performed processes. The results of the study showed that asset specificity (AS) (H1) and trust in the accounting service provider (TRUST) (H5) play important roles in accounting outsourcing. However, the frequency of accounting tasks (FAT) (H4) has been identified as an important factor influencing accounting outsourcing. At the same time, environmental uncertainty (H2) and behavioural uncertainty (UB) (H3) are also important factors in accounting outsourcing. This leads to the conclusion, as in the case of Belgium, that Montenegro is a relatively stable market, and these two factors are not as important here as they are in emerging markets. However, one control variable (gender) played an important role in accounting outsourcing (female managers are less likely to outsource accounting).

**Author Contributions:** Conceptualization, I.T. and S.Đ.; methodology, S.Đ.; software, N.A.; validation, N.A., L.W. and V.K.; formal analysis, V.K.; investigation, S.Đ.; resources, N.A.; data curation, L.W.; writing—original draft preparation, N.A.; writing—review and editing, V.K.; visualization, N.A.; supervision, I.T.; project administration, V.K.; funding acquisition, I.T. All authors have read and agreed to the published version of the manuscript.

**Funding:** This research received no external funding.

**Informed Consent Statement:** Informed consent was obtained from all subjects involved in the study.

**Data Availability Statement:** The data presented in this study are available on request from the corresponding author.

**Conflicts of Interest:** The authors declare no conflict of interest.

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
