# Peer review of "Factors Influencing Accounting Outsourcing Using the Transaction Cost Economics Model"

_ijfs, doi:10.3390/ijfs11020061_

Round 1

Reviewer 1 Report

The main problem of this paper is that it fails to highlight the novelty or additional knowledge it can bring to the literature in a comparison with previous studies in the field. There are quite a number of studies that also look at the factors or the determinants of the accounting outsourcing and also used the TCE approach. The choice of the determinants that the paper makes is also similar to the previous studies. The only significance I can point out is the analysis applied to Montenegro. However, how analysis focusing on Montenegro can shed a light on the new angle that we cannot find in previous studies? What is special about firms in Montenegro?

Some minor comments:

* The introduction is poorly written with no clear motivation, discussion of previous findings or highlight the contribution of the paper. This makes an impression that this paper is just an exercise that applies the same methodology as previous studies to the different data sample of Montenegro.

* Data part is also poorly described as it is lack of how the data is processed/cleaned/prepared. Besides, a correlation matrix among independent variable should be provided and discussed to address the concern of multicollinearity problem.

* The model (1) is wrongly presented - a linear regression model cannot be written in that form, please write up the full model with associated parameters and error term. 

* Results need to be interpreted in a relation with relevant previous studies to highlight the similarities and difference. After that economic rationales should be given.

* There is clear distraction to the reader showing a careless work. For example, the following statement appears in the body of the paper: "Authors should discuss the results and how they can be interpreted from the perspective of previous studies and of the working hypotheses. The findings and their implications should be discussed in the broadest context possible. Future research directions may also be highlighted." - this also an issue that the authors need to address carefully.

Author Response

Dear reviewer,

Thank you very much for your valuable advice and comments. Tried to fix it as much as possible.

Reviewer 2 Report

1. Put * next to the corresponding author in the author list. 

2. Line 37-40: this statement needs to be sustained by at least one citation

3. Line 45: here, the authors should state the aim of the research

4. The Theoretical framework and the problem are well-defined and clear. 

5. Consistency is required all over the paper (choose either Transaction Cost Economics; or, Transaction cost economics.

6. The methodology part is clear; thus, I recommend adding more citations in the paragraphs situated between lines 132-152

7. The Transaction Cost Economy abbreviation was defined at the beginning of the paper; there is no need to explain it again (i.e., line 202).

8. For table 1, please provide a more relevant caption. Tables and figures should stand alone. 

9. For the tables, please add, as a table note, how many enterprises were investigated (n). There are so many tables authors can put the data from table 6 in a graph, for example. 

10. Please ensure a professional figure for figures 1 and 2. Looks like a draft. 

11. In table 5, please make it clear that the variables "University degree" and  "University degree from economics" is regarding the company manager (if so). 

12. Please add as a table note all the abbreviations used in the table. 

13. Discussion section needs to be improved with relevant discussions (the authors need to compare their results with the ones already published by other researchers). 

14. I recommend adding a conclusions and shortcomings section. 

Author Response

(The authors gave the same response as above.)

Round 2

Reviewer 1 Report

I thank the authors for addressing my concerns